# Tweezepy: A Python package for calibrating forces in single-molecule video-tracking experiments

Ian L. Morgan[1]*, Omar A. Saleh[1,2]*

**1** BMSE Program, University of California, Santa Barbara, California, United States of America, **2** Materials Department, University of California, Santa Barbara, California, United States of America

* ilmorgan@ucsb.edu (ILM); saleh@ucsb.edu (OAS)

**Data Availability Statement:** The Tweezepy Python package is available through Github (https://github.com/ianlmorgan/tweezepy), the Python package index (https://pypi.org/project/Tweezepy/), and the Zenodo database

## Abstract

Single-molecule force spectroscopy (SMFS) instruments (e.g., magnetic and optical tweezers) often use video tracking to measure the three-dimensional position of micron-scale beads under an applied force. The force in these experiments is calibrated by comparing the bead trajectory to a thermal motion-based model with the drag coefficient, $\gamma$, and trap spring constant, $\kappa$, as parameters. Estimating accurate parameters is complicated by systematic biases from spectral distortions, the camera exposure time, parasitic noise, and least-squares fitting methods. However, while robust calibration methods exist that correct for these biases, they are not always used because they can be complex to implement computationally. To address this barrier, we present Tweezepy: a Python package for calibrating forces in SMFS video-tracking experiments. Tweezepy uses maximum likelihood estimation (MLE) to estimate parameters and their uncertainties from a single bead trajectory via the power spectral density (PSD) and Allan variance (AV). It is well-documented, fast, easy to use, and accounts for most common sources of biases in SMFS video-tracking experiments. Here, we provide a comprehensive overview of Tweezepy's calibration scheme, including a review of the theory underlying thermal motion-based parameter estimates, a discussion of the PSD, AV, and MLE, and an explanation of their implementation.

## 1 Introduction

Single-molecule force spectroscopy (SMFS) instruments are powerful tools with a wide variety of experimental applications. They can be used to study polymer elasticity [1, 2] and dynamics [3], measure bond energies and lifetimes [4, 5], assess the activity of molecular motors [6, 7], and characterize protein and nucleic acid folding [8].

To obtain accurate and reproducible results, an essential first step in any SMFS experiment is force calibration. Typically, force calibration relies on comparing the thermal motion of a trapped bead to a model derived from the Langevin equation [9]. These methods have limitations; notably, at times, $t \lesssim 10^{-4}$ s, the standard Langevin equation does not account for certain

(https://doi.org/10.5281/zenodo.4948229). All data and analysis files are available from the Zenodo database (https://doi.org/10.5281/zenodo.4948542).

**Funding:** This work was supported by the National Science Foundation under Grant No. 1715627. The funders had no role in study design, data collection and analysis, decision to publish, or preparation of the manuscript.

**Competing interests:** The authors have declared that no competing interests exist.

hydrodynamic effects between the bead and the surrounding fluid [10]. Nevertheless, for longer times, these hydrodynamic effects can be ignored and the bead motion is well-described by the overdamped Langevin equation, which only depends on two parameters: the drag coefficient of the bead, $\gamma$, and the spring constant of the trap, $\kappa$, from which the force can be calculated.

In practice, analyzing and fitting the bead trajectory must be done carefully. Several factors, including spectral distortions, the exposure time of the detection system (e.g., video cameras), parasitic noise (e.g., tracking errors and mechanical drift), and biased fitting, can all lead to inaccurate parameter estimates [11–15]. Robust calibration methods that account for all of these factors exist, yet they can be complex to implement computationally, leading some researchers to opt for alternative strategies [16].

Existing force-calibration software packages [17–20] only account for some sources of bias; most notably they do not account for the finite exposure time of the camera in video-tracking experiments. Thus, it is often up to researchers to write with their own calibration code, of which published examples are only available in proprietary programming languages (e.g., MatLab [21] and LabView [22]), hindering easy access. As has been argued elsewhere [23], different computational implementations, even those based on the same algorithms, can often lead to different numerical outcomes. This lack of standardized calibration methods and computational implementations hinders reproducibility and makes comparison across different research groups, instruments, and experiments difficult [24].

To help improve and standardize SMFS force calibration, we present Tweezepy: a Python package for calibrating forces in single-molecule video-tracking experiments. Tweezepy uses maximum likelihood estimation (MLE) to estimate parameters, and their uncertainties, from a user-provided bead trajectory via a thermal motion-based model of the power spectral density (PSD) or Allan variance (AV). It accounts for the most common sources of biases and parasitic noise in SMFS video-tracking experiments. Moreover, it is written in Python, a popular and freely available programming language, and includes documentation (https://tweezepy.readthedocs.io) with installation instructions and tutorials. It is designed for ease-of-use, only requiring a few lines of simple code, yet it has a versatile object-oriented framework that can be part of a larger scripted workflow or in a Jupyter notebook as a lab journal page [25, 26]. It is developed on GitHub (https://github.com/ianlmorgan/tweezepy) and available through the Python package index, making it easy to distribute and install. The latest stable versions are also archived on the Zenodo database [27].

In this article, we provide a comprehensive overview of Tweezepy's force calibration scheme. In Section 2, we describe several common force calibration methods and motivate the use of the PSD and AV. In Section 3, we give closed-form expressions that account for common sources of parameter biases and parasitic noise in video-tracking experiments, such as the finite exposure time of the detection system and tracking errors. In Section 4, we review how to compute the experimental PSD and AV from a bead trajectory. In Section 5, we describe how to use MLE to reduce biased fitting and estimate parameters and their uncertainties. We note here that estimating parameter uncertainties using MLE has received relatively little attention in the SMFS literature. For experienced readers that are familiar with force calibration theory, we recommend skipping ahead to Section 6, which covers the computational implementation of the calibration methods in Tweezepy. In Section 7, we use Tweezepy to calibrate simulated bead trajectories, and show that it accurately estimates parameters and their uncertainties, similar to previously published results [15].

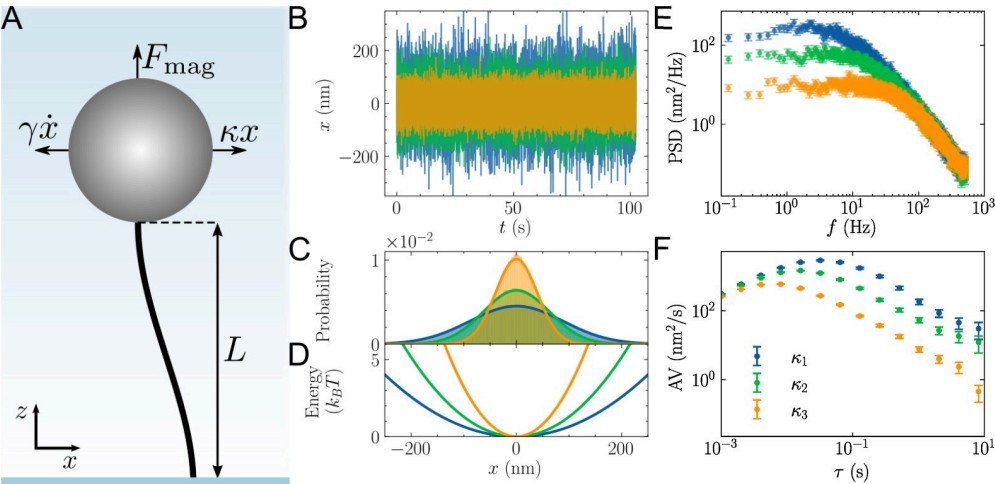

**Fig 1.** (A) Illustration of a magnetic tweezer SMFS experiment in which a polymer is tethered between a glass surface and a paramagnetic bead. Collisions with water molecules drive the bead away from its equilibrium position, creating a restoring force $\kappa x$ from the magnetic trap and a drag force $\gamma \dot{x}$ from the solution. The polymer bead system is treated as an inverted pendulum, such that the upward force, $F_{\text{mag}}$, is determined from the height of the bead above the surface, $L$, and the spring constant, $\kappa$, in the $x$-direction: $F_{\text{mag}} = \kappa L$ [28]. (B) Simulated random, diffusive motion of the bead's $x$ position over time, $t$, with a drag coefficient, $\gamma = 8.38 \times 10^{-6}$ pNs/nm, and three different spring constants: $\kappa_1 = 5.3 \times 10^{-4}$ pN/nm (blue), $\kappa_2 = 1.1 \times 10^{-3}$ pN/nm (green), and $\kappa_3 = 2.6 \times 10^{-3}$ pN/nm (orange). (C) The bead positions follow a Gaussian distribution due to the (D) harmonic potential generated by the applied magnetic trap. (E) The power spectral density (PSD) and (F) Allan variance (AV) of the bead trajectories in (B).

## 2 Background

In a typical SMFS experiment, a polymer is tethered between a surface and a trapped bead (Fig 1A), and the polymer extension is measured while force is applied to the bead. The force on the bead is not known *a priori* and needs to be calibrated.

Most force calibration methods fall into two categories: methods that calibrate against known forces, such as Stokes drag or sedimentation [29], and methods that calibrate based on the thermal motion of the bead [9]. The first category generally relies on intrinsic parameters of the system (e.g. the density and viscosity of the solution) that can be difficult to measure and often vary within an experiment, leading to large uncertainties [30].

In comparison, thermal motion-based calibration methods are advantageous because they only rely on the temperature of the system, which is much easier to measure and control in most experiments. These methods model the trap as a harmonic potential in which the bead undergoes random, diffusive motion (Fig 1B–1D). The applied force is determined from the spring constant of the trap, $\kappa$, and displacement of the bead, $x$, via Hooke's law ($F = -\kappa x$) [13, 31]. By the equipartition theorem, the spring constant of the trap can be related to the standard variance of the bead position, $\sigma_x^2$:

$$\sigma_x^2 = \frac{k_B T}{\kappa}. \tag{1}$$

where $k_B$ is the Boltzmann constant and $T$ is the absolute temperature of the system [32]. While Eq 1 can theoretically give an accurate estimate of $\kappa$ when the time between measurements, $\tau_s$, is much faster than the relaxation time of the bead, $\tau_c \equiv \gamma/\kappa$, in practice, sources of parasitic noise always increase the variance, leading to systematic underestimates of the apparent spring constant [13].

A better approach to thermal motion-based calibration is the PSD, which permits separation of thermal motion from parasitic noise [13, 33]. The PSD describes the distribution of the variance (i.e., total power) across different frequency components in a signal (Fig 1E). Invariably, parasitic noise sources have spectral signatures that differ from those of the bead's thermal motion. As discussed in detail below, when using the PSD to calibrate video-tracking experiments, one needs to account for several factors, including 1) distortions from aliasing and spectral leakage [11, 34], 2) low-pass filtering from the exposure time of the camera [12], and 3) biased parameter estimates from improperly using least squares fitting routines with experimental PSD values that do not have Gaussian-distributed errors [14].

An alternative means of thermal motion-based calibration, that also distinguishes parasitic from thermal noise, is the AV. The AV measures the noise in the bead position over different observation times and was designed as a means of measuring drift in a system [35] (Fig 1F). It was originally introduced into the SMFS literature to assess optimal measurement times [36] and low-frequency noise [37, 38]; however, it was quickly realized that the AV could be directly used for force calibration through fitting [15]. As discussed in detail below, the AV is naturally suited to video-tracking experiments because it intrinsically accounts for low-pass filtering from the exposure time of the camera. However, as with the PSD, improperly using least-squares fitting routines on AV values that do not have Gaussian-distributed errors will lead to biased parameter estimates [15].

When used properly, both the PSD and AV will give accurate parameter estimates under optimal conditions. One of these conditions is that $\tau_s$ should be less than $\tau_c$; further, $\tau_c$ should be less than the total measurement time, $\tau_m$: $\tau_s < \tau_c < \tau_m$ [28]. Hence, assuming $\tau_s$ and $\gamma$ are constant, it is possible to perform calibrations when $\kappa < \gamma/\tau_s$, so long as the measurement time is long enough. Thus, calibration will be easier when $\kappa$ is small, corresponding to small forces or measurements of longer (and thus, more flexible) polymers.

When identifying and accounting for various sources of parasitic noise, the PSD and AV have complementary strengths [38]. The PSD is excellent at identifying high frequency coherent noise sources, such as line frequencies from power sources, while the AV is ideal for identifying low frequency noise sources, such as mechanical drift. In combination, the PSD and AV can be used to identify most forms of parasitic noise. When working with a new or modified instrument, both should be used to determine sources of parasitic noise, which can then be removed, or accounted for in the final fitting procedure.

## 3 Modeling thermal motion in the PSD and AV

### 3.1 Langevin dynamics

Thermal motion-based calibration methods rely on Langevin dynamics, which model the trapped bead in an SMFS experiment as randomly diffusing in a harmonic potential. Collisions between the bead and water molecules create a stochastic (Langevin) force, $F_L$, that obeys the fluctuation-dissipation relation, $\langle F_L(t + t')F_L(t) \rangle = 2\gamma k_B T\delta(t')$, where $\delta(t)$ is the Dirac delta function. In video-tracking SMFS experiments, the bead's motion is well-described by the overdamped Langevin equation [15]:

$$\kappa x(t) + \gamma \dot{x}(t) = F_L(t). \tag{2}$$

## 3.2 A closed-form expression for the PSD

The predicted PSD, $P$, of the bead trajectory at each frequency, $f$, follows from Fourier analysis of Eq 2:

$$P(f) = \frac{k_B T}{2\pi^2 \gamma \left[ \left( \frac{\kappa}{2\pi\gamma} \right)^2 + f^2 \right]}.$$

(3)

For frequencies above the corner frequency, $f_c \equiv \kappa/2\pi\gamma$, the bead motion is purely diffusive and the PSD can be approximated as $P(f) \approx \frac{k_B T}{2\pi^2 \gamma f^2}$. For frequencies below the corner frequency, the bead is constrained by the trap, and the PSD can be approximated as $P(f) \approx \frac{2k_B T \gamma}{\kappa^2}$.

Eq 3 does not account for the exposure time of the camera, $\tau_0$, which introduces a low-pass filter to the experimental bead positions. The predicted PSD that accounts for the exposure time, $P_A$, includes a correction function, $I$ [12]:

$$P_A(f) = P(f)I(f).$$

(4)

where

$$I(f) = \frac{\sin^2(\pi f \tau_0)}{(\pi f \tau_0)^2}.$$

(5)

Eq 4 also needs to be adjusted for aliasing distortions: for an instrument with a sampling rate, $f_s \equiv 1/\tau_s$, the PSD at each positive frequency, $f$, ($0 < f < f_s/2$) contains the summed power of other frequencies, $nf_s$, for all integers, $n$ [11]. The predicted PSD, $P_{A,B}$, that accounts for both the exposure time of the camera and aliasing distortions is given by,

$$P_{A,B}(f) = \sum_{n=-\infty}^{\infty} P_A(|f + nf_s|).$$

(6)

In the special case that $\tau_0 = \tau_s$, the sum in Eq 6 can be performed analytically to give an exact, closed-form expression for $P_{A,B}$ [15]:

$$P_{A,B}(f) = \frac{2k_B T \gamma}{\kappa^3} \left( \kappa + \frac{2\gamma f_s \sin^2\left(\frac{\pi f}{f_s}\right) \sinh\left(\frac{\kappa}{\gamma f_s}\right)}{\cos\left(\frac{2\pi f}{f_s}\right) - \cosh\left(\frac{\kappa}{\gamma f_s}\right)} \right).$$

(7)

Most modern video cameras are designed to maximize captured light, with a dead time ($\sim 10^{-6}$ s) that is much less than the sampling time ($\tau_s \sim 10^{-1}$ s to $10^{-4}$ s). This ensures that the exposure time is about the same as the sampling time, $\tau_0 = \tau_s$, i.e., zero dead-time, fitting the criteria for applying Eq 7.

While sources of parasitic noise will vary among different SMFS instruments, most video-tracking experiments have a frame-to-frame tracking error arising from the imprecision of the bead localization algorithm. Assuming the tracking error is Gaussian-distributed with a standard deviation, $\epsilon$, this adds a frequency-independent white noise term to $P_{A,B}$ [39]:

$$P_{A,B,C}(f) = P_{A,B}(f) + \frac{\epsilon^2}{f_s}.$$

(8)

In the PSD, the effect of tracking errors is most apparent at high frequencies, where the thermal motion is diminished (Fig 2A).

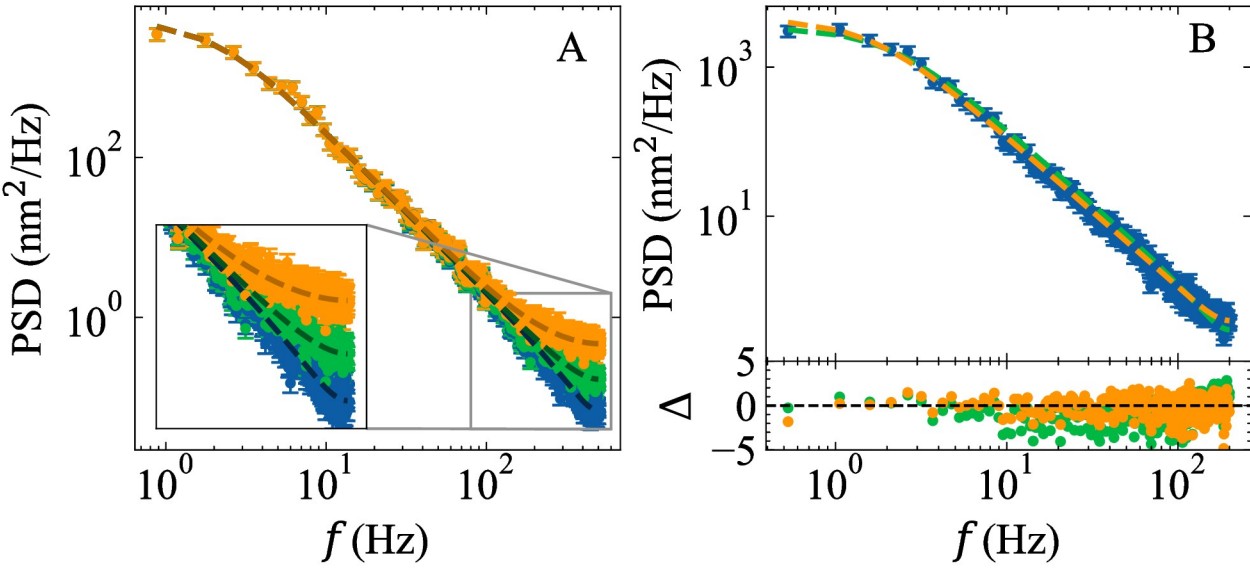

**Fig 2. Example plots of PSDs with per-frame tracking errors.** (A) Simulated bead trajectories with tracking errors lead to deviations in the PSD at high frequencies. The data points correspond to bead trajectories with per-frame tracking errors, $\epsilon = 0$ (blue), 10 (green), and 20 (orange) nm. All trajectories contained $N_x = 20480$ points and were simulated with $f_s = 1000$ Hz, $\gamma = 1.77 \times 10^{-5}$ pNs/nm, and $\kappa = 1.2 \times 10^{-4}$ pN/nm. Dotted lines are overlays of Eq 8 using known parameter values. (B) MLE fits of $P_{A,B}$ (Eq 8, orange dotted line) and $P_{A,B,C}$ (Eq 8, green dotted line) to experimentally derived PSD values (blue points). The experimental data were collected on a bead tethered to double-stranded DNA (contour length $\approx$ 2.8 um) at 400 Hz on a custom-built magnetic tweezer as described in Ref. [40]. $P_{A,B,C}$ Eq 8) is more consistent with the experimental data as judged by the normalized residuals, $\Delta$, and Akaike Information Criterion (AIC = 334 Eq 8 vs. 739 Eq 7). The best fit parameters for $P_{A,B,C}$ are $\kappa = 1.8 \pm 0.2 \times 10^{-4}$ pN/nm, $\gamma = 1.78 \pm 0.04 \times 10^{-5}$ pNs/nm, and $\epsilon = 8.0 \pm 0.3$ nm. All PSD values were computed using Welch's method (Sect. 4.1) with 35 half-overlapping bins. Error bars represent one standard deviation.

### 3.3 A closed-form expression for the AV

For bead motion, the predicted AV, $\sigma_{AV}^2$, at each observation time, $\tau$, is similarly derived through analysis of Eq 2 [15]:

$$\sigma_{AV,A}^2(\tau) = \frac{2k_BT\gamma}{\kappa^2\tau}\left(1 + \frac{2\gamma}{\kappa\tau}e^{-\frac{\kappa\tau}{\gamma}} - \frac{\gamma}{2\kappa\tau}e^{-\frac{2\kappa\tau}{\gamma}} - \frac{3\gamma}{2\kappa\tau}\right).$$

(9)

For observation times that are shorter than the bead relaxation time, $\tau \ll \tau_c \equiv \gamma/\kappa$, neighboring positions are highly correlated, and the AV increases as $\sigma_{AV,A}^2 \approx 2k_BT\tau/3\gamma$. For observation times that are longer than the bead relaxation time, neighboring positions become uncorrelated, and the AV decreases as $\sigma_{AV,A}^2 \approx 2k_BT\gamma/\tau\kappa^2$ [37]. The peak of the transition between the two regimes can be numerically calculated as $\tau_{\max} \approx 1.89\tau_c$ [15].

In its definition, the AV implicitly accounts for the exposure time of the camera and assumes zero dead-time, i.e., $\tau_0 = \tau_s$. As discussed in the previous section, this is usually a reasonable assumption for SMFS video-tracking systems. When this is not the case, the AV is biased and requires an additional correction function [41]. However, conveniently, this bias is negligible when the time between samples is shorter than the bead relaxation time, i.e., $\tau_s < \tau_c$ [37]. Hence, Eq 9 can often be applied without modification for SMFS experiments, regardless of whether dead-time is present [42].

As with the PSD, tracking errors in video-tracking experiments can be accounted for by adding a white-noise term to $\sigma_{AV,A}^2$:

$$\sigma_{AV,A,B}^2(\tau) = \sigma_{AV,A}^2(\tau) + \frac{\epsilon^2\tau_s}{\tau}.$$

(10)

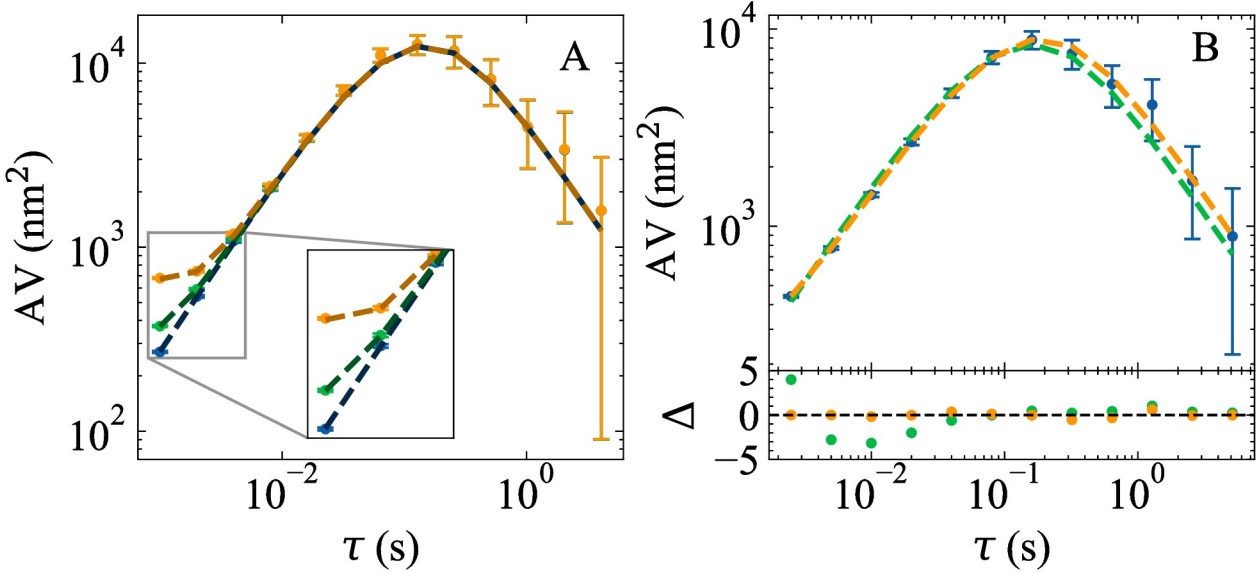

**Fig 3. Example plots of AV values for data with per-frame tracking errors.** (A) Simulated bead trajectories with tracking errors lead to deviations in the AV at short observation times. The data points correspond to the simulated bead trajectories with $\epsilon = 0$ (blue), 10 (green), and 20 (orange) nm. The simulated trajectories are the same as in Fig 2A. Dotted lines are overlays based on Eq 10 with the known parameter values. (B) MLE fits of $\sigma^2_{AV,A,B}$ (Eq 10, dotted orange line) and $\sigma^2_{AV,A}$ (Eq 9, dotted green line) to experimentally derived AV values (blue points). The data are the same as in Fig 3B. $\sigma^2_{AV,A,B}$ (Eq 10) is more consistent with the experimental data as judged by the normalized residuals, $\Delta$, and AIC (AIC = 161 Eq 10 vs. 198 Eq 9). The best fit parameters for $\sigma^2_{AV,A,B}$ are $\kappa = 1.7 \pm 0.1 \times 10^{-4}$ pN/nm, $\gamma = 1.77 \pm 0.04 \times 10^{-5}$ pNs/nm, and $\epsilon = 7.9 \pm 0.9$ nm. Error bars represent one standard deviation.

The effect of tracking errors is most apparent at short observation times, when the bead motion is mostly diffusive (Fig 3).

## 4 Computing the PSD and AV

The SMFS experiment generates an experimental bead trajectory containing $N_x$ points. This trajectory must be converted into a noise metric (the PSD or AV) containing $N_y$ points, which is then fit with the expressions in Section 3 so as to extract parameter estimates. The conversion of the experimental trajectory to the noise metric has a few subtleties which are described here.

### 4.1 Computing the experimental PSD with Welch's method

The experimental PSD values are optimally computed using Welch's method [43]. This method consists of splitting the trajectory into half-overlapping bins, each containing $m$ points. The total number of PSD values and bins are thus $N_y = m$ and $M = 2N_x/m − 1$, respectively. A smaller $m$ improves the signal-to-noise ratio of the final experimental PSD, at the cost of reduced sensitivity at lower frequencies [14]. Each bin consists of bead positions, $\hat{x}_j$ for $j \in (0, 1, 2, \ldots, m − 1)$, and the discrete Fourier transform of each bin is calculated for the frequencies, $f_k = kf_s/2m$ for $k \in (1, 2, \ldots, m)$, as

$$\hat{P}_n(f_k) = \frac{1}{mf_s} \left\| \sum_{j=0}^{m-1} w_j \hat{x}_j \exp\left( -\frac{2\pi ijk}{m} \right) \right\|^2. \tag{11}$$

Note that, in practice, most computational implementations compute the discrete Fourier

transform using a fast Fourier transform algorithm (e.g., the Cooley-Tukey algorithm [44]).
The PSD values of each bin are then averaged together by frequency:

$$\hat{P}(f_k) = \frac{1}{M}\sum_{n=1}^{M}\hat{P}_n(f_k). \tag{12}$$

The windowing function, $w_j$, accounts for the phenomenon of spectral leakage: the finite dura-
tion of the measurement causes power at one frequency to show up at other frequencies [45].
Most computational implementations of Welch's method use the Hann windowing function
[46],

$$w_j = \sqrt{\frac{8}{3}}\sin^2\left(\frac{\pi j}{b}\right), \tag{13}$$

which reduces the total power of each experimental PSD value in a frequency-independent
manner, which is then corrected by the leading factor of $\sqrt{8/3}$. The use of the Hann window,
in conjunction with half-overlapping bins, means that data near the termini of one bin is
diminished by the window, but that same data is near the center of the next bin, and thus cap-
tured by the window; this provides a reasonable trade-off between over- and under-utilizing
all of the data [43]. Occasionally, after performing Welch's method, the calculated PSD values
are logarithmically binned to help visualize power-law behavior [11].

## 4.2 Computing the overlapping AV

The experimental AV is optimally computed from the bead trajectory by partitioning it into
octave-sampled, overlapping bins [35]. Octave sampling consists of using bin lengths, $m_k$, in
powers of 2, i.e., $m_k = 2^k$ for $k \in (1, \ldots, N_y)$, where $N_y = \lfloor\log_2(N_x/2)\rfloor$. The bin lengths deter-
mine the number of overlapping bins, $M = N_x - 2m_k + 1$, and the observation times, $\tau = m_k\tau_s$,
where $\tau_s$ is the sampling time. For each $\tau$, the experimental AV, $\hat{\sigma}_{AV}^2$, is calculated as one-half
the mean-squared difference of consecutive average bin positions:

$$\hat{\sigma}_{AV}^2(\tau) = \frac{1}{2(M-1)}\sum_{n=1}^{M-1}(\bar{x}_{n+1} - \bar{x}_n)^2 \tag{14}$$

where $\bar{x}_n$ is the average of bead positions, $\hat{x}_j, j \in (1, 2, \ldots, m_k)$:

$$\bar{x}_n = \frac{1}{m_k}\sum_{j=1}^{m_k}\hat{x}_j. \tag{15}$$

In practice, computing all the average bin positions for each $\tau$ can be slow, so an equivalent,
but more computationally efficient, method is often used [41, 47].

## 5 Biased fitting

After computing the set of experimental AV or PSD values, $\hat{y}_k, k \in (1, 2, \ldots, N_y)$, they are com-
pared to the Langevin model predictions, $y_k$ (Eqs 7–10), using maximum likelihood estimation
(MLE), to extract the best-fit parameter estimates for $\gamma$ and $\kappa$. MLE accounts for the expected
probability distributions of each experimental value. For the AV and PSD, the probability, $p_k$,
of measuring each experimental value is given by the Gamma probability distribution

function:

$$p_k(\hat{y}_k, y_k(\gamma, \kappa)) = \frac{\hat{y}_k^{\eta_k - 1} e^{-\hat{y}_k/\theta_k}}{\theta_k^{\eta_k} \Gamma(\eta_k)} \qquad (16)$$

where $\eta_k$ is termed the shape parameter, $\theta_k = y_k/\eta_k$ is termed the scale parameter, and $\Gamma$ is the gamma function.

For the PSD, the shape parameter is given by the number of bins, $\eta_k = M$, which is notably the same for all values $\hat{y}_k$. For the AV, the shape parameter is generally $\eta_k = \nu_{AV,k}/2$, where $\nu_{AV,k}$ counts the degrees of freedom for each value. $\nu_{AV,k}$ depends on the number of differences used to calculate the $k^{th}$ value, as well as the dominant type of noise at that value [41]. It is common to approximate $\nu_{AV,k}$ from the number of successive differences between non-overlapping bins of length $m_k$ that are present in the trajectory, $\nu_{AV,k} = (N_x/m_k) - 1$ [15]; however, this is an underestimate.

For both the PSD and AV, as $\eta_k \to \infty$, the Gamma distribution approaches a normal (Gaussian) distribution, and least-squares fitting can be used. However, for moderate values of $\eta_k$, the distribution is not normal, and least-squares fitting routines lead to biased parameter estimates. While it is possible to correct for these biases analytically, in general, MLE gives more accurate parameter estimates [14].

## 5.1 Maximum likelihood estimation

MLE is based on estimating the parameters, $\hat{\gamma}$ and $\hat{\kappa}$, that maximize the likelihood function, $L$, which is the joint probability of all $p_k$:

$$L(\gamma, \kappa) = \prod_{k=1}^{N_y} p_k(\hat{y}_k, y_k(\gamma, \kappa)). \qquad (17)$$

In practice, rather than maximizing $L$, it is more convenient to minimize the cost function, $\ell \equiv -\ln L$. Given Eqs 16 and 17, the cost function is given by:

$$\ell(\gamma, \kappa) = \sum_{k=1}^{N_y} \eta_k \left[ \frac{\hat{y}_k}{y_k(\gamma, \kappa)} + \ln(y_k) \right] + \text{const}, \qquad (18)$$

where the final term is a constant with respect to the parameters. Minimizing $\ell$ is a straightforward optimization problem that can be solved numerically with standard algorithms (e.g., Nelder-Mead [48]).

## 5.2 Parameter uncertainties

After finding the best-fit parameters, $\hat{\gamma}$ and $\hat{\kappa}$, an estimate of their uncertainties can be found from standard approaches: In particular, the likelihood function, $L$, is assumed to have a Gaussian form in the vicinity of its maximum. Then, the matrix of second partial derivatives of $L$ (i.e., the Hessian matrix) are calculated, and inverted to find the squared uncertainties (i.e., the covariance matrix). Details of this approach can be found in statistical references, e.g. Ref. [49].

The applicability and robustness of Hessian-based estimates of parameter uncertainty rests on whether $L$ behaves as a Gaussian over a significant region near $(\hat{\gamma}, \hat{\kappa})$. This question is distinct from that of the proper distribution governing the AV or PSD estimates themselves (i.e, the values $\hat{y}_k$)– the $\hat{y}_k$ values, in certain cases, are calculated from a relatively small number of samples, and so are distributed in a highly non-Gaussian manner (Eq 16), which drives the use

of MLE rather than least-squares optimization methods. However, the MLE cost function is based on a relatively larger number of points ($N_y$), and so, by the central limit theorem, is well-modeled as Gaussian. Therefore, in practice, the Hessian approach typically results in robust estimates of parameter uncertainty.

That said, in some cases, it may not be appropriate to approximate the likelihood function as a Gaussian, e.g., when there are small sample sizes, outliers, or complex parameter correlations. Such situations can be handled by an alternate, numerical approach in which a Monte Carlo algorithm is used to sample the parameter space [49].

To carry out Monte Carlo sampling, several 'walkers' are initiated around the estimated parameters. These 'walkers' take random steps in parameter space and evaluate the cost function, which determines whether each step is accepted or rejected. After a predetermined number of steps, a histogram of the accepted steps is used to generate an empirical probability distribution for the parameters (Fig 4). From this distribution, the confidence intervals can be

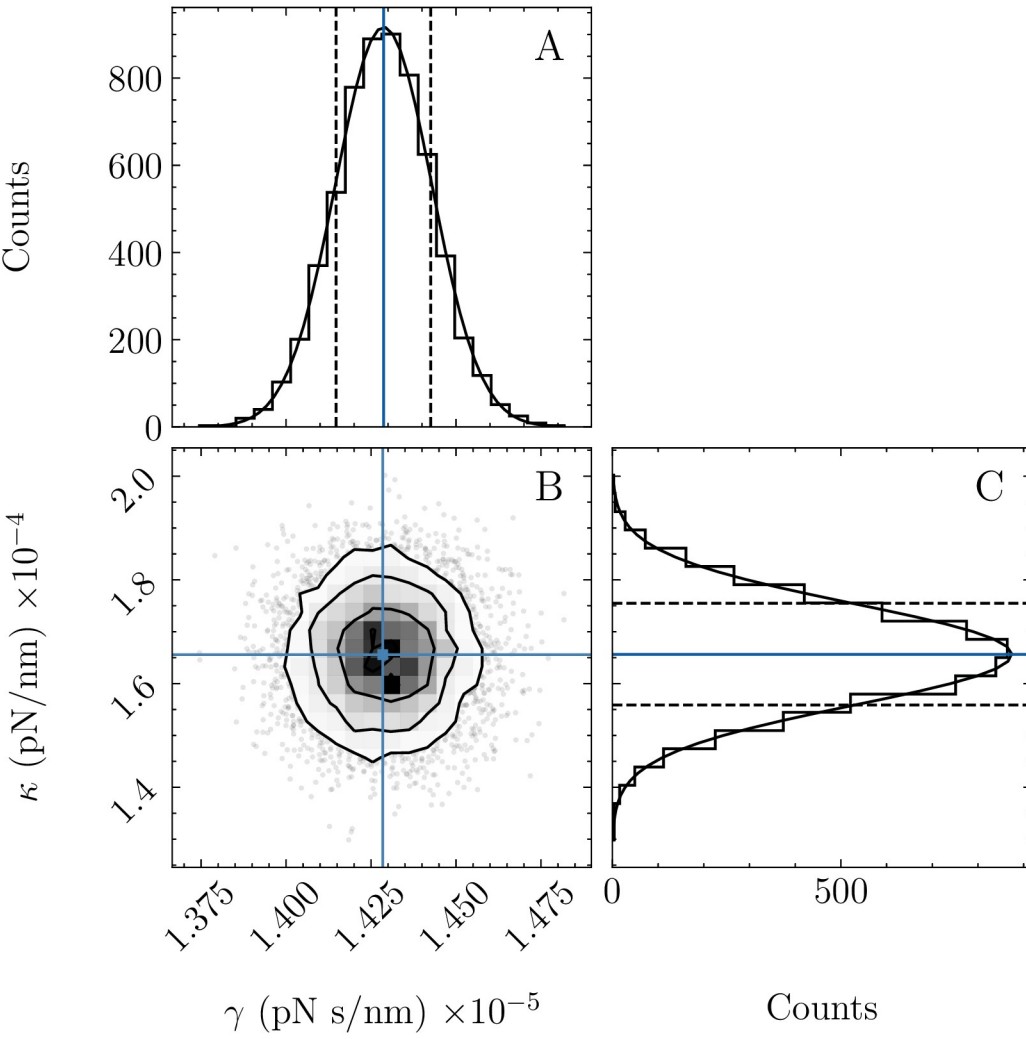

**Fig 4. Histograms of accepted steps from MCMC sampling of experimental data in Fig 2B.** One-dimensional histograms of (A) $\gamma$ and $\kappa$ values with best-fit parameters (blue line) and 15.8th and 84.2nd percentiles (black dotted lines). The histograms follow a Gaussian distribution (black solid line) as assumed in the Hessian method. Two-dimensional histogram of (B) $\gamma$ and $\kappa$ values with best-fit parameters (blue lines). The contour (black) lines are the one, two, and three standard deviations. Points outside of three standard deviations are plotted individually.

evaluated. Typically, the standard errors are estimated as half the difference between the 15.8th and 84.2nd percentiles, which corresponds to one standard deviation for a Gaussian distribution.

## 5.3 Fit quality

After fitting, the quality of the fit needs to be judged. There are several means of judging the quality of the fit, each with its own advantages and disadvantages. The simplest means is to look at the normalized residuals, $\Delta_k$, i.e., the deviations between the experimental and predicted values:

$$\Delta_k = \frac{\hat{y}_k - y_k(\hat{\gamma}, \hat{\kappa})}{\sigma_{y,k}}, \tag{19}$$

where $\sigma_{y,k}$ is the standard deviation of the $k^{\text{th}}$ experimental value. The normalized residuals can be plotted to assess systematic deviations between the data and the fit.

If the normalized residuals follow a Gaussian distribution, their variance corresponds to the reduced chi-squared value, $\chi^2_{v_y}$:

$$\chi^2_{v_y} = \frac{\chi^2}{v_y} = \frac{1}{v_y}\sum_{k=1}^{N_y} \Delta_k^2. \tag{20}$$

The degrees of freedom, $v_y$, are estimated as $v_y = N_y - K$, where $K$ is the number of fit parameters [49]. A reduced chi-squared value of one is usually considered a 'good' fit [49]. A reduced chi-squared value that is greater than one is generally considered a 'poor' fit, whereas a reduced chi-squared value that is less than one is usually considered an overfit. However, the reduced chi-squared value has a variance that scales as $2/v_y$, so values based on small sample sizes or models with a large number of parameters can be misleading.

Instead, the cumulative distribution function of chi-squared-distributed values, $F$, is usually a better measure of fit quality (also termed the support for the fit) [17]:

$$F(\chi^2, v) = \frac{1}{\Gamma(v_y/2)} \int_0^{\chi^2/2} z^{v_y/2-1} \exp(-z)dz. \tag{21}$$

The support evaluates the probability that repeating the experiment will give a larger $\chi^2_{v_y}$ value. It is closely related to the p-value, i.e., $1 - F$. For a 'good' fit, the support is expected to be close to one.

While the support for the fit evaluates agreement between the experimental and predicted values, other statistical metrics, such as the Akaike Information Criterion (AIC), are better at comparing models with different numbers of parameters [50]. The AIC balances the quality of fit with the number of parameters. It is calculated as

$$\text{AIC} = 2K - 2\ln(\hat{L}). \tag{22}$$

Due to varying constants and sample sizes, individual AIC values are not informative. Instead, the data are considered to be best described by the model with the lowest AIC value, $\text{AIC}_{\text{min}}$, regardless of the number of parameters, when the difference between two models' AIC values is $\Delta_{\text{AIC}} = \text{AIC} - \text{AIC}_{\text{min}} \geq 4$ [50], as applied in Figs 2 and 3.

## 6 Tweezepy

Tweezepy is a Python package for thermal motion-based force calibration in SMFS video-tracking experiments that estimates parameters and their uncertainties from a user-provided bead trajectory, using MLE, via the PSD or AV. For a detailed explanation of the package, including expected inputs and outputs, the reader is referred to the docstrings and usage examples. In this section, we discuss specific implementation choices and practical considerations for using the package.

To use Tweezepy, the user provides a bead trajectory and sampling frequency to either the PSD or AV class objects. Given this information, Tweezepy computes the experimental values and compares them to a user-selected predictive model using MLE. After fitting, it reports the parameter estimates and uncertainties, as well as the fit quality. The experimental and predicted values, as well as the normalized residuals, can be visualized using the included plotting functions.

To compute the experimental PSD, Tweezepy uses Welch's method (Sec. 4.1). By default, it uses a Hann windowing function with three half-overlapping bins. The signal-to-noise ratio of the experimental PSD values can be improved by increasing the number of bins, which helps to visualize the values and slightly reduces the parameter uncertainties. However, there is a trade-off: as the number of bins increases, the low-frequency resolution decreases. For low corner frequencies, this can lead to a substantial bias in the parameter estimates (Fig 5). Unfortunately, it is difficult, *a priori*, to know the optimal number of bins, so it is up to the user to choose the appropriate number of bins. This is a drawback of the PSD method.

Tweezepy uses MLE to compare the experimental PSD values to, by default, Eq 7, which accounts for both aliasing and the finite bandwidth of the detection system. This function assumes the exposure time is the same as the time between measurements, i.e., zero dead-time. As discussed in Section 3.2, this assumption is typically good for video-tracking experiments. When dead-time is present in the measured bead trajectory, the user can also select an alternative function that uses a closed-form expression based on Eq 6 that assumes a negligible exposure time [14, 15], i.e., it only accounts for aliasing. Additionally, the user can select to use a modified version of either function that includes tracking errors from video-tracking bead localization algorithms (e.g., Eq 8).

To compute the experimental AV, Tweezepy uses the octave-sampled, overlapping approach as described in Sec. 4.2. It empirically determines the degrees of freedom for each value using the Greenhall algorithm [51], based on the dominant type of power-law noise for each experimental value. It estimates the dominant type of noise using the Lag1 autocorrelation algorithm [52]. This algorithm has lower precision for AV values with fewer bins, i.e., for long observation times when $N_x/m_k < 32$. Typically, this corresponds to the three or four AV values with the longest observation times. For these points, Tweezepy assumes the dominant noise type is the same as the last point that satisfies $N_x/m_k > 32$. If the algorithm fails to estimate any of the dominant types of noise, it warns the user and falls back on using the approximate degrees of freedom based on nonoverlapping bins, i.e., $v_{AV} = N_x/m_k - 1$. The user can choose to use only the approximate degrees of freedom by setting the keyword argument 'edf' to 'approx'. For visualization purposes, the user can select to plot all or decade-spaced observation times. As discussed in Section 4.2, the approximate degrees of freedom give nearly identical parameter estimates but underestimate the confidence for each AV value, leading to slightly larger parameter errors. After computing the experimental AV values, Tweezepy compares them to Eq 9. Additionally, the user can select a predefined function that accounts for tracking errors from the video-tracking bead localization algorithms (Eq 10).

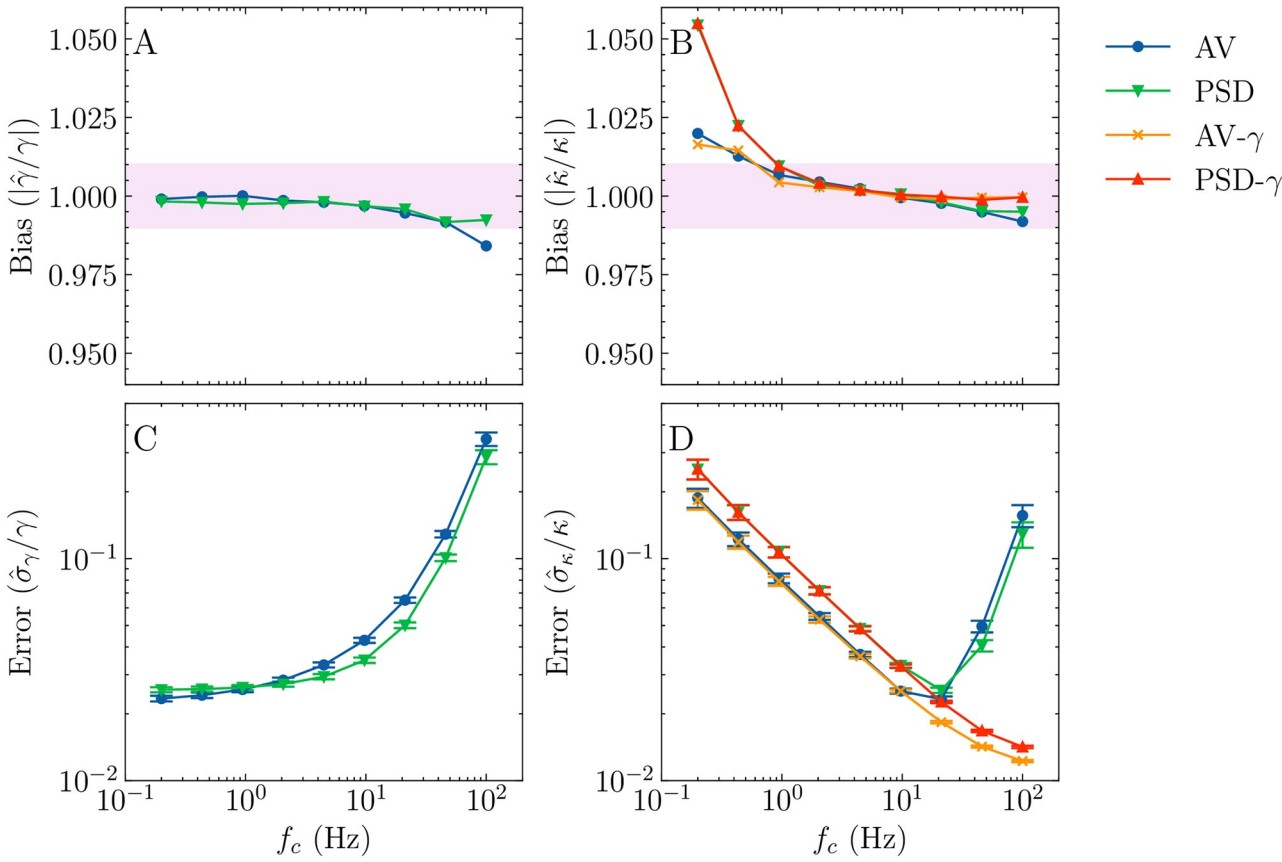

**Fig 5. Bias and error for the AV and PSD methods in Tweezepy.** (A and C) Bias in estimated parameters as compared to known parameter values for (A) $\gamma$ and (B) $\kappa$. (C and D) Error in estimated parameters using Hessian method for (C) $\gamma$ and (D) $\kappa$. Fixing $\gamma$ reduces the bias and error in $\kappa$ at high corner frequencies by removing parameter correlations. Each point represents the median of 1000 simulations; each simulation contained 4096 bead positions with a constant drag coefficient, $\gamma = 1 \times 10^{-5}$ ps/N, and sampling frequency, $f_s = 100$ Hz. The corner frequency was varied logarithmically between 0.2 Hz to 100 Hz. The blue and green points represent two-parameter AV and PSD method fit results. The orange and red points represent fixed gamma AV and PSD method fit results. In the bias plots, the magenta box represents the ±1% bias region.

In addition to its predefined functions, Tweezepy also accepts user-defined functions to compare to the experimental values. If these functions include additional fitting parameters, it is recommended that they are compared to a function without the additional parameters using the AIC to avoid overfitting (Sec. 5.3). Additionally, the normalized residuals can be plotted and visualized to detect deviations between the data and theoretical values. Typically, it is easier to visualize the residuals of the AV compared to the PSD because it has fewer values.

Evaluating the AIC can also be useful for determining whether one or more parameters is poorly constrained during the fit. As discussed later (Sec. 7), in some cases, the sampling frequency is not fast enough to resolve the purely diffusive motion of the bead, causing $\gamma$ to be poorly constrained during fitting. However, $\kappa$ can usually still be reliably estimated by fixing $\gamma$ to a known value. Tweezepy contains keyword arguments for fixing any of the parameters for its predefined functions during the fit. Ideally, the known $\gamma$ value should be estimated from the same bead at a lower force, and adjusted for surface effects using Faxen's correction [53]. To determine whether fixing a parameter is necessary, the AIC of the fits with and without fixing are compared, and the fit with the lowest AIC value is used.

When sources of parasitic noise are present but cannot be properly described by the selected model analytically, it is recommended that the user subtract a reference spectrum or

bandpass filter the measured data. For example, mechanical drift in experiments often manifests as $1/f$ power-law noise at low frequencies in the PSD or $\tau$ power-law noise at long observation times in the AV [38]. By excluding the regions of the spectrum where drift (or other sources of noise) dominates during fitting, the parameters can still be accurately estimated. In Tweezepy, the user can select upper and lower cutoff frequencies (or observation times) to compare the function to a limited range of the spectrum using the keyword argument 'cutoffs'.

To calculate parameter uncertainties, Tweezepy evaluates and inverts the expected Hessian (Sec. 5.2). To evaluate the Hessian, it uses the Autograd Python package. Autograd uses automatic differentiation to evaluate derivatives by repeatedly applying the chain rule to elementary operations. This speeds up code and reduces numerical precision errors that can occur with numerical and symbolic differentiation [54, 55]. In addition to calculating and inverting the Hessian, Tweezepy contains an optional method for robust uncertainty estimates via Monte Carlo sampling (Sec. 5.1). This method uses the Emcee Python package [56] to carry out Monte Carlo sampling. In our hands, this more robust method, but slower (with computation time on the order of 10s), produces Gaussian parameter distributions (Fig 4) and nearidentical uncertainty estimates to the faster method ($\approx$10 ms) that inverts the Hessian. For example, for the data in Fig 4, using the Hessian method gives $\gamma = 1.43 \pm 0.02 \times 10^{-5}$ pNs/nm and $\kappa = 1.6 \pm 0.1 \times 10^{-4}$ pN/nm, while the MCMC method gives $\gamma = 1.43 \pm 0.01 \times 10^{-5}$ pNs/nm and $\kappa = 1.6 \pm 0.1 \times 10^{-4}$ pN/nm.

In addition to the packages mentioned above, Tweezepy makes use of the standard python library [57], including NumPy [58], SciPy [59], and Numba [60]. All the package dependencies are noted in the requirements and setup files for easy installation.

## 7 Results

To evaluate Tweezepy, we sought to benchmark its fit results against known parameter values. Following the example of Ref. [15], we simulated bead trajectories using $N_x = 4096$, $f_s = 100$ Hz, and $\gamma = 1.0 \times 10^{-5}$ pNs/nm (a typical drag coefficient for a one micron spherical bead in water), and varied the corner frequency, $f_c$, logarithmically from 0.2 Hz to 100 Hz, giving spring constants $\kappa$ that ranged from $1.4 \times 10^{-4}$ pN/nm to $6.8 \times 10^{-3}$ pN/nm. To carry out the simulations, we iteratively generated successive bead positions, without tracking errors, from Eq 2 [61] (Fig 1B). To mimic the effects of the camera exposure time, we used a time step of $\delta t = 1/(1000f_s)$, split the trajectory into bins of 1000 points, and took the average of each bin to generate a downsampled trajectory. For each corner frequency, we simulated 1000 trajectories. For each trajectory, we computed and fit the PSD to Eq 7 (Fig 2B) and the AV to Eq 9 (Fig 3B) using Tweezepy to estimate the parameters and their uncertainties. To estimate bias, we calculated the ratio of the median parameter estimates and true values. To estimate the error, we calculated the ratio of the median parameter uncertainties and true values.

For nearly all corner frequencies, the bias for $\gamma$ and $\kappa$ estimates is within ±1% (Fig 5A and 5C magenta box). There is an increase in the bias and error for $\kappa$ estimates at lower corner frequencies because, for the simulated length of the trajectory, the bead motion is mostly unconstrained by the trap. As a result, the $\kappa$ estimate is poorly constrained during fitting. This effect is slightly worse for the PSD because binning decreases its low frequency resolution more than the AV. In practice, this bias can usually be reduced by increasing the length of the trajectory [62].

At high corner frequencies, $f_c \gtrsim f_s/8$, there is an increase in the error and a slight bias in both parameters (Fig 5C and 5D), consistent with previous findings [14, 15]. This is because the sampling frequency is not fast enough to resolve the unconstrained diffusive bead motion, which leads to poorly constrained $\gamma$ estimate. The correlations between the $\gamma$ and $\kappa$ parameters

lead to a poorer $\kappa$ estimate. This is why it is advantageous to collect bead trajectories for force calibration in SMFS video-tracking experiments at the highest available sampling frequency.

It is worth noting that the authors in Ref. [21] recommend using a low-pass-corrected standard variance calibration method [12] to avoid the small bias at high corner frequencies with the PSD and AV. However, we note that their implementation of this alternative method fixes $\gamma$ to a known value during fitting, removing the parameter correlations. We find that fixing $\gamma$ with the PSD and AV similarly removes the increase in the error and slight bias for $\kappa$ at high corner frequencies (Fig 5C and 5D). This suggests that, under optimal conditions, all three methods can accurately estimate parameters.

## 8 Conclusions

In this article, we have reviewed robust thermal motion-based force calibration in SMFS experiments using the PSD and AV, and discussed implementing them computationally into a Python package, Tweezepy, that is freely available on Github and the Python package index.

In designing Tweezepy, our goal was to make it as robust, versatile, and user-friendly as possible. It uses MLE to estimate parameters via the PSD or AV, and goes beyond previous computational implementations by calculating the empirical degrees of freedom for the overlapping AV and determining parameter uncertainties from MLE, either by inverting the Hessian or, optionally, via Monte Carlo sampling. It includes several predefined closed-form expressions that account for the most common biases and parasitic noise in SMFS video-tracking experiments. Yet, it also accepts user-defined functions, so it can be adapted to account for additional sources of noise or applied to other problems that rely on fitting the PSD or AV of a bead trajectory, e.g., torque calibration [42]. Lastly, Tweezepy uses sensible default options to make it easy-to-use, only requiring a few straightforward lines of code, with computation times on the order of 10 ms. Our hope is that Tweezepy can serve as a useful tool to improve and standardize force calibration across different SMFS research groups, instruments, and experiments.

## Acknowledgments

We thank Frank Truong and Sarah Innes-Gold for helpful discussions and beta testing Tweezepy.

## Author Contributions

**Conceptualization:** Ian L. Morgan.

**Data curation:** Ian L. Morgan.

**Funding acquisition:** Omar A. Saleh.

**Methodology:** Ian L. Morgan.

**Resources:** Omar A. Saleh.

**Software:** Ian L. Morgan.

**Supervision:** Omar A. Saleh.

**Validation:** Ian L. Morgan.

**Visualization:** Ian L. Morgan.

**Writing – original draft:** Ian L. Morgan.

**Writing – review & editing:** Ian L. Morgan, Omar A. Saleh.

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
