## [Decision Letter · Decision Letter 0]

2 Aug 2021

PONE-D-21-19568

Tweezepy: A Python package for calibrating forces in single-molecule video-tracking experiments

PLOS ONE

Dear Dr. Morgan,

Thank you for submitting your manuscript to PLOS ONE. After careful consideration, we feel that it has merit but does not fully meet PLOS ONE’s publication criteria as it currently stands. Therefore, we invite you to submit a revised version of the manuscript that addresses the points raised during the review process.

Please consider the comments made by the reviewers, which are partly aimed at further improving the usabilitiy of your work. The comments particularly aim at making the software more easily accessible for new users of the techniques. In this context, you may also want to consider publishing a detailed documentation of the software or providing a graphical user interface.

We look forward to receiving your revised manuscript.

Kind regards,

Kerstin G. Blank

Academic Editor

PLOS ONE

Journal Requirements:

"We thank Frank Truong and Sarah Innes-Gold for helpful discussions and beta testing 445

Tweezepy. This work was supported by the National Science Foundation under Grant 446

No. 1715627."

"This work was supported by the National Science Foundation under Grant No. 1715627."

Reviewers' comments:

Reviewer's Responses to Questions

**Comments to the Author**

1. Is the manuscript technically sound, and do the data support the conclusions?

Reviewer #1: Yes

Reviewer #2: Yes

2. Has the statistical analysis been performed appropriately and rigorously? 

Reviewer #1: N/A

Reviewer #2: Yes

3. Have the authors made all data underlying the findings in their manuscript fully available?

Reviewer #1: Yes

Reviewer #2: Yes

4. Is the manuscript presented in an intelligible fashion and written in standard English?

Reviewer #1: Yes

Reviewer #2: Yes

5. Review Comments to the Author

Reviewer #1: Morgan and Saleh present in this paper a python package program which can be used for force calibrations using video tracking in single molecule force spectroscopy. This article is overall well written and the topic well explained. It reviews both PSD and Allan variance methods for achieving this purpose, and provides an extensive description of how to use these methods for calibration. Even though this topic has been extensively described in the past, the authors offer an automated package of programs that might be of interesting use in general in the field to achieve the calibration of their instrument and would allow users, especially novice ones, to achieve this purpose as precisely as possible and taking into account different kind of parameters of such a system. The article also describes various sources of noise and errors in the measurement and describe how to correct them. One interesting feature and strength of this program is that it offers different robust methods to be able to fit as accurately as possible their data and also provides the users with automated robust methods to estimate biases and errors of their fit.

However there are points which I feel are missing or need to be addressed better in this article and the accompanying software, therefore I suggest the following minor revisions:

- In the introduction the authors discuss forces and calibration, could they provide a short description relating this calibration to an actual molecule as an example where they show forces and the parameters they want to fit (γ and κ) in for example a small illustration? Can they explain which molecules are better to be used experimentally for calibration purposes using their methods ?

- Can they discuss for which force regimes it is better to use PSD or Allan variance?

- In line 113- 118, the authors mention complementarity of PSD and Allan variance, can they explain how to combine them in terms of force calibration?

- In eq 4, the authors should mention that P_A (f) is the experimentally measured Power spectrum.

- In line 202: is there a difference in terms of results for the PSD using Welch’s or the FFT Cooley-Tukey algorithm?

- Are the simulated traces mentioned in section 6 and 7 the same than in fig 1 and 2? If yes please refer it in text.

- Line 341: can authors explicit what they mean by “previously estimated noise type” is it from previous points in AV?

- In line 410: authors say “this bias can be reduced by increasing the length of the trajectory” can authors show this improvement in their figures? Why are only examples of short traces shown?

- Line 411 onward: there is an unclear description of the minimum in bias and error in relation to figure. As it is supposed to be the case for both parameters, why are only C and D mentioned? There also seem to be no clear minima in the bias for γ as well as its error for f_c >f_s/8 which looks more like an increasing tendency. Is this observation only valid for κ? In this case it is not very consistent with the rest of the description.

In general the figures and their legends are also not very clear and I suggest the following edits:

- Fig 1 and 2: some confusion and mixtures in text that needs to be looked at. Before describing figures A and B separately authors should explicit what orange blue and green plot correspond to in general, this can also be mentioned in the figure where they could add the ϵ parameter values next to each plot color. “AIC= value vs value”, authors could explain for which eq vs eq each value corresponds. Titles of both these figures could be changed to more general description as this figure in my opinion shows more than tracking errors but is an example of PSD, Allan variance and comparison of fits.

- Fig 3: mark panels A, B, and C for consistency and easier description in legend for reader to better understand. Confusing histogram κ on bottom right which seems to show κ = f(κ). Flip this graph 90 deg. to the right so it coincides with axis label of the 2D histogram. Add units to γ and κ in the axis labels.

- Fig 4: Description in legend difficult to follow, use lettering to describe the graphs. Describe better what are the plots in function of and explicit that corner frequency is f_c. Reorder sentences for better understanding of legend as well.

With regard to the actual software

After reviewal of the content of the program Tweezepy, I would suggest the following modifications :

- Either creating an interface for such a program, which might be of interest for users who are not advanced in programming or python scripting and this might facilitate the general use of Tweezepy.

- Or to write more detailed documentation for users. The documentation I found ist not sufficient to make sure that any data obtained from instruments can be read in readily. Many of the parameters that need to be fed into the program are to my finding not so straightforward. As an example, the trace data has to be in nanometer per second for the MLE fit function to work (error message without an error description encountered otherwise). Explain this more detailed in the documentation what to feed into the various functions that the users might need for their calibration.

Otherwise non-expert might be quickly uninterested and experts might not use the software because they have their own.

In conclusion I would recommend publishing this paper after these revisions in the content of the article have been addressed.

Reviewer #2: Morgan and Saleh have developed a Python based analysis package, “ Tweezepy,” to extract the stiffness and drag coefficient from single-molecule thermal noise trajectories. The package uses both power-spectral density (PSD) and Allan variance (AV) approaches to extract the stiffness and drag coefficient from the position fluctuations of the single-particle captured in the input data. The authors extend both the PSD and AV approaches to account for typical camera-based detection limitations having to do with finite sampling time, aliasing, and additional noise sources in the analytical descriptions used as the basis of fitting the data. They apply Maximum Likelihood Estimation based fitting, and importantly provide two means of estimating the uncertainties in the fitted parameters. The mathematical and analytical approaches are sound and the authors have done a good job in providing an overview of the critical steps in the process along with appropriate citations to the relevant literature. The examples are illustrative and the authors provide useful guidelines for improving the accuracy of the extracted parameters. I envision that this will be a useful tool for many single-molecule biophysicists who are interested in extracting the stiffness and drag coefficient of single-particle manipulation measurements. I have a few minor points that the authors should consider prior to publication.

1. P 2, L 82-83: The statement regarding the insensitivity of thermal calibration approaches to the intrinsic parameters of the system is not strictly correct. This is correct for the simple variance based approach in equation 1, but not generally true since power spectral methods require accurate measures of the solution viscosity, the particle size, and the position of the particle relative to surfaces.

2. P5, L169-171: this sentence is confusing: ” Conveniently this bias is negligible when τs τc [37], so Eq. 9 can usually be applied, without further modification, to photodiode-based detection systems that include dead-time [44]” Typically one thinks of photodiode systems to be faster and exhibit no dead-time in comparison to camera based systems. Perhaps this could be better described based on the discrepancy between the effective bandwidth of the detector and the sampling rate – which leads to an effective dead-time perhaps. Without more details this statement seems incorrect and perhaps could be rephrased to get at the underlying issue.

3. P6 L182. Based on issues related to noise estimation and improving the robustness of the fitting, Flyvbjerg and coworkers, e.g. reference 11, have suggested binning the PSD results in a similar manner as the AV results. Can the authors comment on the choice to not bin the PSD in light of the suggestion that this improves fitting for optical trapping experiments?

4. Figure 2 legend: “know” should be “known”

5. Figure 3. The 2-D histogram has a mix of large “pixels” toward the center of the distribution but small point that lie outside the last contour line – can the authors explain these discrepant points?

6. For the data in figure 3 it would be helpful to compare this particular case to a Gaussian approximation. Or compare the uncertainties derived through the MC simulation process to those estimated from a Gaussian distribution.

7. Figure 4. It seems surprising that the bias in the stiffness is worse for the PSD model in which the drag coefficient is fixed at low cut-off frequencies. I would expect that the fit should improve, or be no worse, when the known drag coefficient is used in the fitting function. Can the authors comment on this somewhat surprising result?

8. The authors do not specifically address drift in the measurements. It would be useful to include a brief comment on how drift would manifest in the AV and PSD treatments of the data and how it could be included I the fitting or eliminated from consideration.

6. PLOS authors have the option to publish the peer review history of their article (what does this mean?). If published, this will include your full peer review and any attached files.

Reviewer #1: No

Reviewer #2: No

---

## [Author Response · Author response to Decision Letter 0]

29 Oct 2021

Please see the cover letter and reviewer response attached for an in-depth response to the reviewer and editor comments. 

In addition, we have uploaded the figures as .tif files to comply with the PLOS ONE's figure requirements. We have also removed the funding statement from the acknowledgements section. We intend to use the current Funding statement "This work was supported by the National Science Foundation under Grant No. 1715627."

---

## [Decision Letter · Decision Letter 1]

1 Dec 2021

PONE-D-21-19568R1Tweezepy: A Python package for calibrating forces in single-molecule video-tracking experimentsPLOS ONE

Dear Dr. Morgan,

Thank you for submitting your manuscript to PLOS ONE. After careful consideration, we feel that it has merit but does not fully meet PLOS ONE’s publication criteria as it currently stands. Therefore, we invite you to submit a revised version of the manuscript that addresses the points raised during the review process. Please submit your revised manuscript by Jan 15 2022 11:59PM. If you will need more time than this to complete your revisions, please reply to this message or contact the journal office at plosone@plos.org. Please include the following items when submitting your revised manuscript:A rebuttal letter that responds to each point raised by the academic editor and reviewer(s). You should upload this letter as a separate file labeled 'Response to Reviewers'.A marked-up copy of your manuscript that highlights changes made to the original version. You should upload this as a separate file labeled 'Revised Manuscript with Track Changes'.An unmarked version of your revised paper without tracked changes. You should upload this as a separate file labeled 'Manuscript'.If applicable, we recommend that you deposit your laboratory protocols in protocols.io to enhance the reproducibility of your results. Protocols.io assigns your protocol its own identifier (DOI) so that it can be cited independently in the future. For instructions see: https://journals.plos.org/plosone/s/submission-guidelines#loc-laboratory-protocols. Additionally, PLOS ONE offers an option for publishing peer-reviewed Lab Protocol articles, which describe protocols hosted on protocols.io. Read more information on sharing protocols at https://plos.org/protocols?utm_medium=editorial-email&utm_source=authorletters&utm_campaign=protocols.

We look forward to receiving your revised manuscript.

Kind regards,

Kerstin G. Blank

Academic Editor

PLOS ONE

Journal Requirements:

Additional Editor Comments:

Both reviewers agree that the manuscript has improved significantly and should be accepted; however, reviewer 1 also suggests a few small corrections that you may want to implement.

Reviewers' comments:

Reviewer's Responses to Questions

**Comments to the Author**

1. If the authors have adequately addressed your comments raised in a previous round of review and you feel that this manuscript is now acceptable for publication, you may indicate that here to bypass the “Comments to the Author” section, enter your conflict of interest statement in the “Confidential to Editor” section, and submit your "Accept" recommendation.

Reviewer #1: All comments have been addressed

Reviewer #2: All comments have been addressed

2. Is the manuscript technically sound, and do the data support the conclusions?

Reviewer #1: Yes

Reviewer #2: Yes

3. Has the statistical analysis been performed appropriately and rigorously? 

Reviewer #1: N/A

Reviewer #2: Yes

4. Have the authors made all data underlying the findings in their manuscript fully available?

Reviewer #1: Yes

Reviewer #2: Yes

5. Is the manuscript presented in an intelligible fashion and written in standard English?

Reviewer #1: Yes

Reviewer #2: Yes

6. Review Comments to the Author

Reviewer #1: Morgan and Saleh have responded to all the questions and addressed all comments in this revision. I feel the manuscript has improved significantly. I particularly appreciate the more detailed documentation for their python program and the associated tutorial section. This allows first time users a much better access to this software, which looks to be very helpful for future biophysicists.

Therefore I would strongly suggest to publish this article.

While reading, I noted a few minor issues, which I would like to list here:

- all axes should be labeled. for histograms, the y-axis is often not labeled (e.g. Fig 4). What about counts?

- for better referencing, I was missing in section 7 a refer to Fig. 1 - e.g. in lines 415/416. This makes it clear to the non-expert reader what are the simulated traces. In the same section, the authors can refer to Fig. 2 and 3 for the PSD and AV.

Reviewer #2: The authors have addressed my concerns and, I think, have addressed the main concerns of reviewer 1. I recommend publication.

7. PLOS authors have the option to publish the peer review history of their article (what does this mean?). If published, this will include your full peer review and any attached files.

Reviewer #1: No

Reviewer #2: No

---

## [Author Response · Author response to Decision Letter 1]

9 Dec 2021

We appreciate the time and effort of the editor and reviewers. We have incorporated the small changes suggested by Reviewer one to the manuscript and marked them in red. These changes are described in more detail in the accompanying reviewer reply.

---

## [Editor Report · Decision Letter 2]

16 Dec 2021

Tweezepy: A Python package for calibrating forces in single-molecule video-tracking experiments

PONE-D-21-19568R2

Dear Dr. Morgan,

We’re pleased to inform you that your manuscript has been judged scientifically suitable for publication and will be formally accepted for publication once it meets all outstanding technical requirements.

Kind regards,

Kerstin G. Blank

Academic Editor

PLOS ONE

---

## [Editor Report · Acceptance letter]

20 Dec 2021

PONE-D-21-19568R2 

Tweezepy: A Python package for calibrating forces in single-molecule video-tracking experiments 

Dear Dr. Morgan:

I'm pleased to inform you that your manuscript has been deemed suitable for publication in PLOS ONE. Congratulations! Your manuscript is now with our production department. 

Kind regards, 

on behalf of

Prof. Dr. Kerstin G. Blank 

Academic Editor

PLOS ONE